# Quaternization-spiro design of chlorine-resistant and high-permeance lithium separation membranes

Huawen Peng[1], Kaicheng Yu[2], Xufei Liu[1], Jiapeng Li[1], Xiangguo Hu ◎[2] & Qiang Zhao ◎[1] ✉

Current polyamide lithium extraction nanofiltration membranes are susceptible to chlorine degradation and/or low permeance, two problems that are hard to reconcile. Here we simultaneously circumvented these problems by designing a quaternized-spiro piperazine monomer and translating its beneficial properties into large-area membranes ($1 \times 2$ m$^2$) via interfacial polymerization with trimesoyl chloride. The quaternary ammonium and spiral conformation of the monomer confer more positive charge and free volume to the membrane, leading to one of the highest permeance (~22 L m$^{-2}$ h$^{-1}$ bar$^{-1}$) compared to the state-of-the-art Mg$^{2+}$/Li$^+$ nanofiltration membranes. Meanwhile, membrane structures are chlorine resistant as the amine–acyl bonding contains no sensitive N-H group. Thus the high performance of membrane is stable versus 400-h immersion in sodium hypochlorite, while control membranes degraded readily. Molecular simulations show that the high permeance and chlorine resistance, which were reproducible at the membrane module level, arise from the spiral conformation and secondary amine structures of the monomer.

Interfacial polymerization has been known for more than half a century to manufacture fibers, capsules and membranes through the condensation of reacting monomers at interfaces of immiscible solvents[1–3]. The condensation of amines and acyl chlorides at water–hexane interfaces leads to the formation of polyamide thin film composite membranes (TFCMs) that are dominating desalination and nanofiltration applications[4–6]. In addition, polyamide membranes are essential in more applications such as gas separation, pervaporation, forward osmosis, and batteries[7–10]. Membrane properties (chlorine resistance[11–13], free volumes[14–16], thickness[17–19], etc.) are affected by the chemical structures of reacting monomers, which should meet multiple requirements, including interfacial diffusivity, condensation reactivity, and structure stability[20–24]. It is an important method to prepare advanced membranes by translating prescribed properties of monomers into corresponding membranes[18,25].

The rapid development of lithium-based energy has boosted the global lithium demand, leading to lithium supply shortage worldwide and the rocketing lithium price in past years[26–28]. Currently, ca. 70 % of lithium is extracted from salt lakes, in which the separation of Mg$^{2+}$ and Li$^+$ (hydration radius: 4.3 vs. 3.8 Å) is one of the main challenges[29–31]. Ion mixtures with close sizes, yet different valences, could be separated by nanofiltration membranes through size sieving and charge repulsion. Thus positive membranes are required for the nanofiltration separation of Mg$^{2+}$/Li$^+$ mixtures[32,33], but commercial polyamide membranes made from piperazine (PIP)–trimesoyl chloride (TMC) reaction show low Mg$^{2+}$/Li$^+$ selectivity due to their negative charge[5,6]. Alternatively, polyethyleneimine (PEI) containing abundant amine groups have been frequently exploited, as the replacement of PIP, to prepare positively charged membranes for Mg$^{2+}$/Li$^+$ separation[34]. Increasing works have been focused on PEI-TMC membranes, aiming to improve Mg$^{2+}$/Li$^+$

[1]Key Laboratory of Material Chemistry for Energy Conversion and Storage, (Ministry of Education), School of Chemistry and Chemical Engineering, Huazhong University of Science and Technology, 430074 Wuhan, P. R. China. [2]National Engineering Research Center for Carbohydrate Synthesis, Jiangxi Normal University, 330022 Nanchang, P. R. China. ✉e-mail: zhaoq@hust.edu.cn

permeance by surface modification, hybridization, etc.[35–39]. However, structures and separation performances of PEI-TMC membranes are not tolerant to chlorine chemicals (e.g., NaClO), widely used as water disinfectants and membrane detergents[12,40,41]. This is because the condensation between primary amine (-NH2) groups (in PEI) and acyl groups (in TMC) yields amide bonds (-CO-NH-) containing active hydrogen atoms that are susceptible to chlorine. The hydrogen of -CO-NH- bond will be attacked by chlorine to form -CO-NCl-, which will be transformed into unstable ring-chlorinated products[42–45]. Such reactions lead to the degradation of PEI-TMC membranes by chlorine oxidants, a longstanding problem that also exists with other membranes derived from primary amine monomers[41,42]. In addition, pristine PEI-TMC membranes suffer from relatively low permeance (~5 L m⁻² h⁻¹ bar⁻¹), though it could be improved by relatively complicated modifications. As such, there is an urgent need for new monomers that confer both chlorine resistance and high permeance to Mg²⁺/Li⁺ separation membranes, an elusive challenge hardly been addressed by far.

Previous studies have shown that chlorine resistance of amide bonds is improved when they contain NO chlorine-active hydrogen (-CO-N-)[40,45,46], such as chlorine-resistant PIP-TMC membranes, despite their low Mg²⁺/Li⁺ separation performance. To this end, reacting monomers should contain secondary amines (-NH) rather than primary amines (-NH2) to eliminate chlorine-sensitive amide bonds in resultant membranes. Meanwhile, positive charges are needed for Mg²⁺/Li⁺ separation, while the conformation of monomers could be contorted to increase the microporosity of resultant membranes. In

consideration of these needs, we are motivated to encode such properties in monomers designed and translate them into chlorine-resistant, high-performance Mg²⁺/Li⁺ separation membranes. A one-step quaternization reaction was exploited to synthesize a quaternized-spiral piperazine (QSPIP) monomer featuring spiral conformation and quaternary/secondary amines. Indeed, the permeance of QSPIP-TMC membranes is ~5 times improved compared to analogous polyamide membranes, and the high separation performance is stable in 400-h continuous tests against NaClO.

## Results and discussion
### Monomer synthesis and membrane characterizations

As shown in Fig. 1a, the QSPIP monomer was synthesized by the cyclization reaction between piperazine (PIP) and bis(2-chloroethyl)amine in water. The QSPIP shows two ¹H NMR peaks (Fig. 1b) at 3.7 ppm (labeled H I) and 4.0 ppm (labeled H II) and two ¹³C-NMR peaks at 37 ppm (labeled C I) and 56 ppm (labeled C II) (Fig. 1c). Area ratio of the two peaks in ¹H NMR is 1.03, in good agreement with QSPIP's structure. A peak at m/z = 156.15 in mass spectroscopy was observed, which is consistent with the chemical structure of QSPIP ($C_8H_{18}N_3^+$, $M_W = 156$, Supplementary Fig. 1). The ratio of carbon to nitrogen elements of QSPIP was measured to be 2.67, close to the theoretical value of 2.69 (Fig. 1d). The positive charge of QSPIP was verified by counter ion exchange experiment (Supplementary Fig. 2). The 3D chemical structure (after energy minimization) of QSPIP shows that QSPIP features spiral conformation and a plane angle of 65.4° (Fig. 1a, right). Noteworthy, the QSPIP has not been exploited for the preparation of separation membranes.

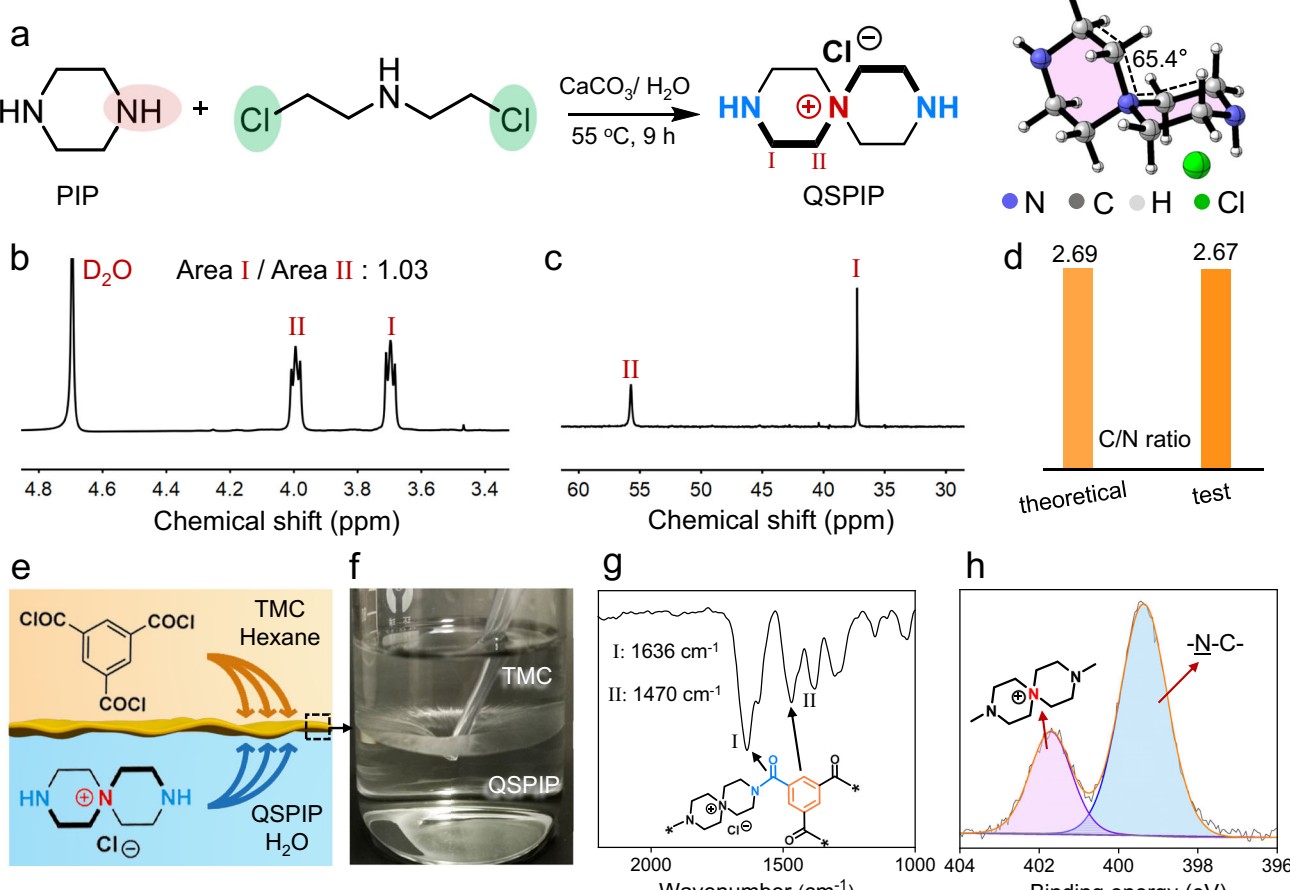

**Fig. 1 | Synthesis and interfacial polymerization of QSPIP. a** Synthesis and 3D conformation of QSPIP. **b** ¹H NMR, **c** ¹³C NMR, and **d** elemental analysis of QSPIP. **e** Schematic and **f** optical image of the interfacial polymerization between QSPIP/ water solution (0.5 wt%) and TMC/hexane solution (0.3 wt%) in a glass beaker. **g** ATR-FTIR and **h** XPS characterizations of the QSPIP-TMC freestanding membrane prepared in (**f**).

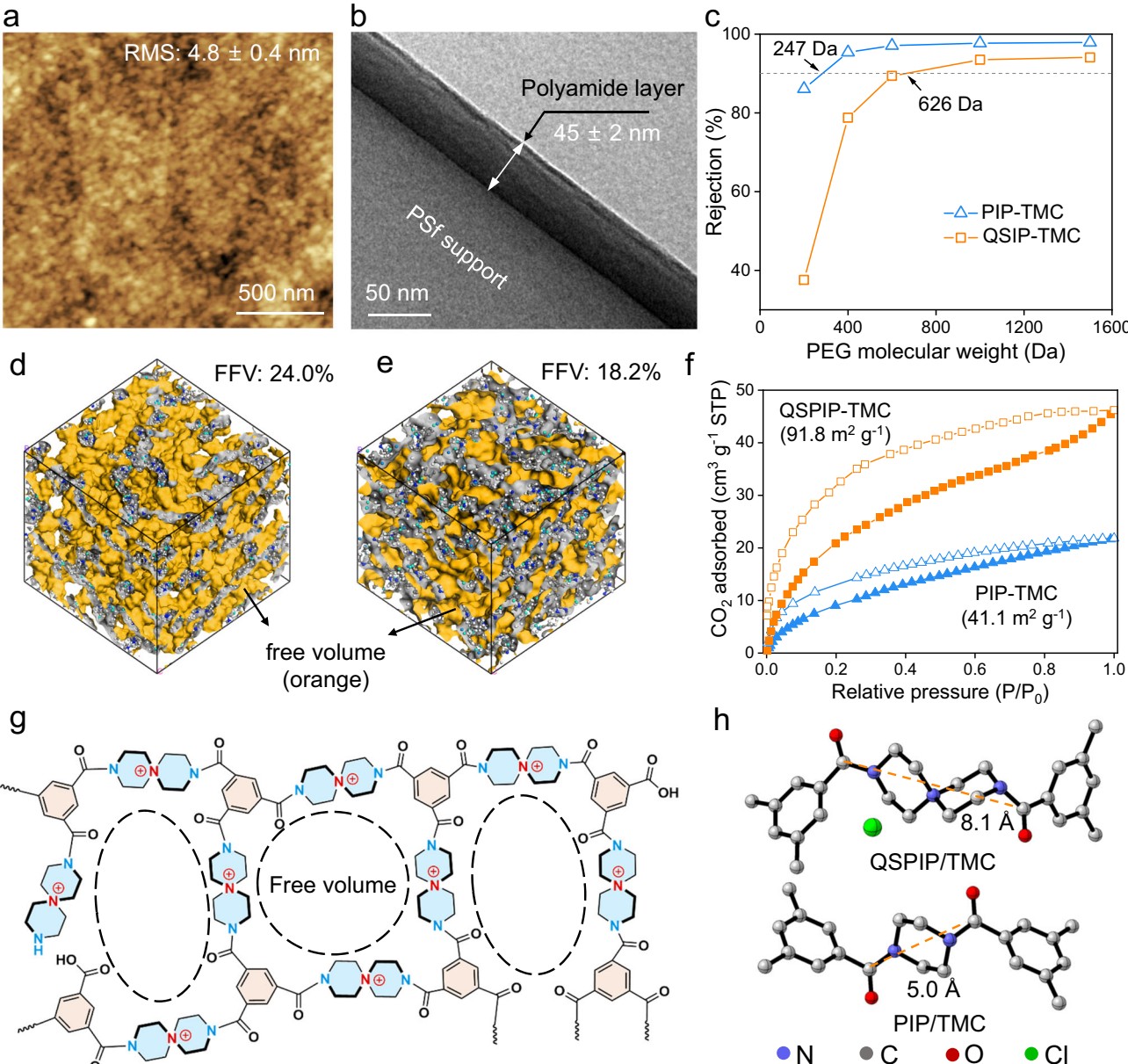

**Fig. 2 | Characterizations of QSPIP-TMC membranes. a** AFM surface morphology and **b** TEM cross-sectional morphology of the QSPIP-TMC membrane. The dark region is QSPIP-TMC due to the density contrast. **c** Rejections of QSPIP-TMC and PIP-TMC membranes toward polyethylene glycol with different molecular weight. **d**, **e** Molecular dynamics simulation of free volume of QSPIP-TMC and PIP-TMC networks, respectively. The orange and gray color indicate the voids of polymer and occupied space of polymer skeleton. **f** $CO_2$ absorption/desorption isothermals of QSPIP-TMC and PIP-TMC. **g** Schematic structures of the QSPIP-TMC network. **h** The 3D structure of QSPIP-TMC and PIP-TMC linkage.

As shown in Fig. 1e, f, the QSPIP−TMC interfacial polymerization was conducted in a glass beaker, in which TMC/hexane solution (0.1 wt %) and QSPIP aqueous solution (0.5 wt%) were brought in contact. The QSPIP monomers diffused across the water–hexane interface, whereby the [2+3] condensation between secondary amine groups (from QSPIP) and acyl chloride groups (from TMC) occurred at the interface, leading to the formation of a thin film. Crosslinking degree of the QSPIP-TMC freestanding membrane is ~72% (Supplementary Fig. 3). Figure 1g shows that the QSPIP-TMC freestanding membrane reveals two characteristic FT-IR peaks at 1636 and 1470 cm$^{-1}$, which are assigned to the amine−acyl amide bond and the vibration of benzene rings from TMC[17]. In addition, the QSPIP-TMC freestanding membrane shows an XPS peak at 401.8 eV that corresponds to quaternary ammonia (-N$^+$) (Fig. 1h)[47].

Then the QSPIP−TMC interfacial polymerization was conducted on surfaces of polysulfone (PSf) supporting membranes to prepare TFCMs for nanofiltration tests. Here, the interfacial polymerization conditions were tuned to optimize the separation performance of the QSPIP-TMC membrane (Supplementary Fig. 4). Both the ATR-FTIR and XPS characterizations confirmed that QSPIP-TMC membranes have been prepared on the PSf supports (Supplementary Fig. 5). Figure 2a shows that the QSPIP-TMC membrane consists of nanoaggregates that are densely packed, and its surface roughness is $4.8 \pm 0.4$ nm. Thickness of the QSPIP-TMC selective layer is $45 \pm 2$ nm (Fig. 2b), thinner than PIP-TMC control membranes which are normally >100 nm under similar conditions[48–50]. QSPIP is a polarized molecule bearing quaternary ammonium charge, and its diffusion rate ($D_{QSPIP} = 1.2 \times 10^{-10}$ m$^2$ s$^{-1}$) in nonpolar hexane solution is slower than that of PIP ($D_{PIP} = 5.8 \times 10^{-10}$ m$^2$ s$^{-1}$, Supplementary Fig. 6), resulting in narrower interfacial reaction region and thinner membranes. In literature, thinner membranes were also obtained by

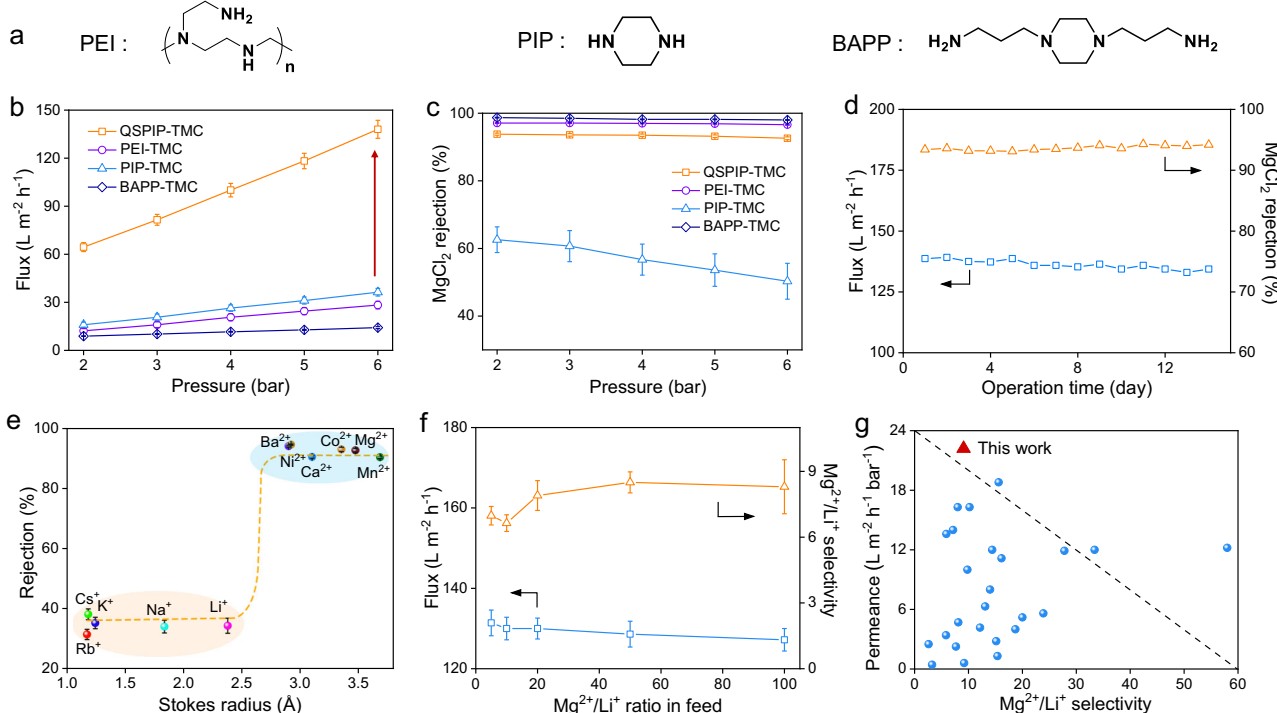

**Fig. 3 | Separation performance of membranes. a** Chemical structures and abbreviations of control monomers exploited to prepare nanofiltration membranes by interfacial polymerization with TMC. Effect of pressure in **b** flux and **c** MgCl$_2$ rejection of QSPIP-TMC, PEI-TMC, PIP-TMC, BAPP-TMC membranes (feed: 1000 ppm MgCl$_2$). **d** Effect of operation time in the nanofiltration performance of QSPIP-TMC membrane (feed: 1000 ppm MgCl$_2$, 6 bar). **e** Ions rejection of QSPIP-TMC membrane for filtrating different ion solutions (feed: 1000 ppm, 6 bar). **f** Effect of Mg$^{2+}$/Li$^+$ ratio in flux and selectivity of QSPIP-TMC membrane at 6 bar. **g** Comparison of QSPIP-TMC membrane with the recent Mg$^{2+}$/Li$^+$ separation membranes operated under cross-flow nanofiltration. The corresponding references in (**g**) are specified in Supplementary Table 1.

reducing the diffusion rate of reacting monomers[22–24]. Moreover, various electrolyte monomers have been studied to prepare nanofiltration membranes, but most of them were employed as a co-monomer in interfacial polymerization or for modification of existing membranes[51–54]. It is noted that QSPIP is one of the rare strong electrolytes[55] exploited as standalone monomers for straightforward preparation of nanofiltration membranes. In addition, the QSPIP-TMC membrane shows good hydrophilicity (water contact angle: ~34°) and antibacterial property toward *S. aureus* (Supplementary Fig. 7).

Figure 2c shows that the molecular weight cut-off and mean pore diameter (Supplementary Fig. 8) of QSPIP-TMC membranes are 626 Da and 6.1 Å, higher than that of PIP-TMC membranes (247 Da, 3.2 Å). As predicted by molecular dynamics (MD) simulation, the fractional free volume (FFV, the orange region) of QSPIP-TMC and PIP-TMC networks is 24.0% (Fig. 2d) and 18.2 % (Fig. 2e), respectively. The FFV value of QSPIP-TMC is close to other nanomembranes prepared by contorted monomers such as triptycene tetra-acyl chloride (Trip)[16]. Meanwhile, the main pore diameter of QSPIP-TMC network derived from molecular simulation is 6–10 Å (Supplementary Fig. 9), larger than that of PIP-TMC network (4−7 Å). Figure 2f shows that the Brunauer-Emmett-Teller specific surface area of the QSPIP-TMC polymer, tested by CO$_2$ adsorption/desorption isotherms at 273 K, are 91.8 m$^2$ g$^{-1}$, 2.2 times as high as that of the PIP-TMC polymer network (41.1 m$^2$ g$^{-1}$). Thus both the characterizations and simulations show that the QSPIP-TMC membrane exhibits loose structures enhancing microporosity. As schemed in Fig. 2g, such loose structures may stem from the ring skeleton of QSPIP with larger sizes and spiral configuration. The distance of two N atoms in the adjacent amide of QSPIP-TMC is 8.1 Å (Fig. 2h), 1.5 times longer than that of PIP-TMC (5.0 Å). Collectively, QSPIP leads to more space facilitating water transportation through the QSPIP-TMC membrane.

## Ion separation performance

The separation performance of QSPIP-TMC was compared with three control membranes made from the interfacial polymerization of TMC with PIP, PEI, and 1,4-Bis(3-aminopropyl)piperazine (BAPP) monomers, respectively (Fig. 3a). PIP and PEI are benchmark monomers widely used to prepare negative and positive polyamide membranes[5], respectively. BAPP is a control monomer containing primary amines (−NH$_2$) instead of secondary amine (-NH). Figure 3b shows that the flux of all membranes is positively correlated to operation pressure, reasonably because larger pressure equals larger driven force for transmembrane pass of water[56]. At 6 bar pressure, the flux of QSPIP-TMC membrane is 138 L m$^{-2}$ h$^{-1}$, which is 4.8, 3.8 and 9.7 times as high as that of PEI-TMC (28 L m$^{-2}$ h$^{-1}$), PIP-TMC (36 L m$^{-2}$ h$^{-1}$) and BAPP-TMC (14 L m$^{-2}$ h$^{-1}$) membranes, respectively. The improved flux of QSPIP-TMC membrane is related to the enhanced microporosity as QSPIP-TMC features the highest FFV (Fig. 2d, e, Supplementary Fig. 10). Figure 3c shows that the MgCl$_2$ rejection (R$_{MgCl2}$) of QSPIP-TMC membrane is ~93% at 6 bar, slightly lower than PEI-TMC (96.6%) and BAPP-TMC membrane (98.0%), but much higher than the R$_{MgCl2}$ of PIP-TMC membrane (50.3%). The higher R$_{MgCl2}$ of the QSPIP-TMC membrane is likely due to the quaternary ammoniums that intensify its positive charge (Supplementary Fig. 11). Moreover, the flux of the QSPIP-TMC membrane is stable during 2 weeks of continuous filtration of MgCl$_2$ solution (Fig. 3d).

Figure 3e shows that the QSPIP-TMC membrane exhibits good rejections (>90%) to divalent cations, e.g., Ba$^{2+}$, Ni$^{2+}$, Ca$^{2+}$, Co$^{2+}$, Mg$^{2+}$, Mn$^{2+}$, while the rejections to single valence cations, e.g., Cs$^+$, Rb$^+$, K$^+$, Na$^+$, Li$^+$ are relatively low (~30%). The flux of QSPIP-TMC toward feeds containing these mono/divalent ions is ~140 L m$^{-2}$ h$^{-1}$ (Supplementary Fig. 12). Figure 3f shows that QSPIP-TMC membrane exhibits good Mg$^{2+}$/Li$^+$ selectivity (~8.7) and ultrahigh permeance (~22 L m$^{-2}$ h$^{-1}$ bar$^{-1}$) when filtrating 100 Mg$^{2+}$/Li$^+$ ratio mixtures. Compared with the state-of-the-art lithium extraction nanofiltration membranes (Fig. 3g), the

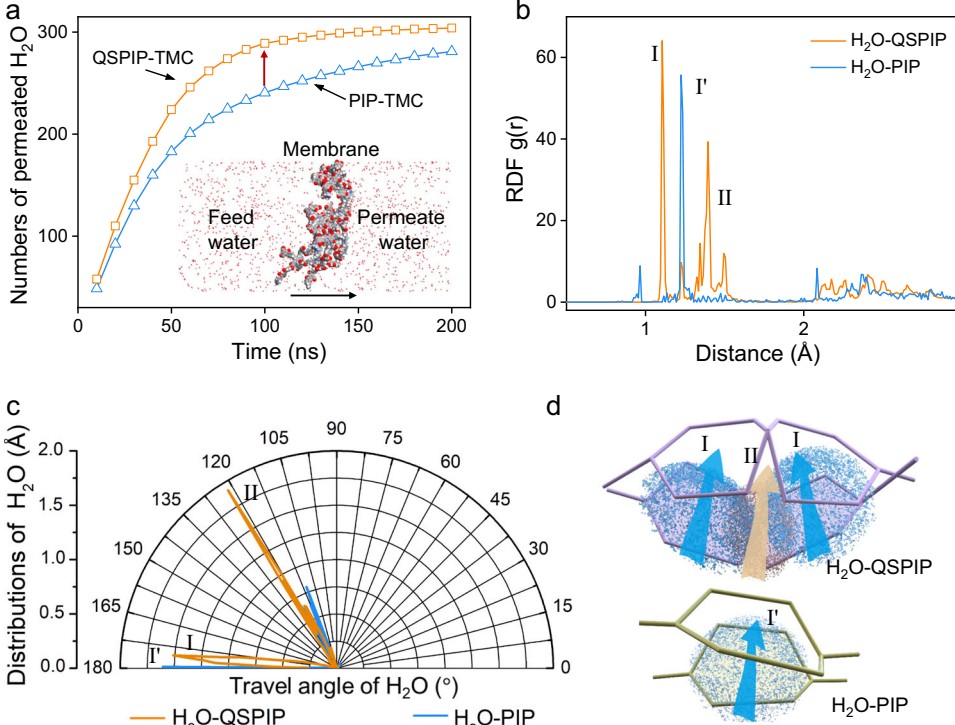

**Fig. 4 | Simulations of water diffusion through the QSPIP/PIP-TMC membrane. a** Effect of time in numbers of water passing through QSPIP-TMC and PIP-TMC membranes. **b** Radial distribution function (RDF) between water molecules and QSPIP (or PIP) moieties in QSPIP-TMC (or PIP-TMC) membranes. **c** The travel angles of water molecules through QSPIP and PIP. **d** Schematic of water transportation through voids produced by QSPIP and PIP.

QSPIP-TMC membrane features a good combination of high permeance and selectivity. In literature, high-permeance $Mg^{2+}/Li^+$ separation membranes were normally prepared by additional modification of pristine PEI-TMC membranes[35–37], requiring multiple preparation steps. Here the QSPIP-TMC membrane was prepared by one-step interfacial polymerization on commercial supports, without the need for surface modifications or intermediate layers. The facile preparation of QSPIP-TMC membranes is beneficial for practical manufacturing and real applications.

Molecular dynamics simulations were conducted to rationalize the mass transport through the QSPIP-TMC membrane. QSPIP (or PIP) and TMC monomers with a mass ratio of 5 were put in a simulation box ($105 \times 105 \times 210$ Å). Both monomers move randomly to facilitate the amine–acyl chloride condensation reaction (Supplementary Fig. 13) and membrane formation[57]. As schemed by the inserted carton in Fig. 4a, 500 water molecules were added in the left chamber, diffused through the membrane, and collected in the right chamber. Figure 4a shows that the total number of water (NW) molecules in the right chamber increases with time and finally equilibrium. The slope of the QSPIP-TMC line is higher than that of the PIP-TMC line, indicating that the rate of water transport in QSPIP-TMC membrane is faster (Fig. 4a). When the running time is ~100 ns (indicated by the red arrow), the NW of QSPIP-TMC membrane and PIP-TMC membrane are 290 and 240, respectively.

The radial distribution function (RDF) values of water in QSPIP-TMC and PIP-TMC membranes during water permeation were studied (Fig. 4b). For the RDF of water around QSPIP, two pronounced peaks are seen at 1.1 Å (I) and 1.4 Å (II), which are due to water appearing around the spiro-piperazine ring (I) and the quaternary ammonium (II), respectively. By contrast, only one sharp peak at 1.2 Å (I') was seen for that of PIP, which indicates water distribution around the planar piperazine. The distance of peak I (1.1 Å) is 0.12 Å smaller than that of peak I' (1.22 Å), which is due to the stronger electrostatic attraction between quaternary ammonium of QSPIP and water molecules. As evidence, water molecules travel through the space of QSPIP

moieties from two main angles at ~123° and ~175° (Fig. 4c), corresponding to paths facilitated by structures II and I. The angle to spiro-piperazine ring (I) ranges from 172° to 180°, while that is limited at 180° for piperazine (I') of $H_2O$-PIP. As schemed in Fig. 4d, more voids were produced by QSPIP in the QSPIP-TMC membrane compared with PIP-TMC due to the spiral conformation of QSPIP. Water molecules permeate across the QSPIP-TMC membrane via both the piperazine plane and region of quaternary ammonium, while water permeates across the PIP-TMC membrane mainly via the piperazine plane. These results signify that there are more alternative water paths in the QSPIP-TMC membrane, i.e., the additional water path due to the spiral, nonplanar conformation of QSPIP.

**Chlorine resistance and mechanisms**

The QSPIP-TMC, PEI-TMC and BAPP-TMC membranes were immersed in NaClO solution (200 ppm, pH = 6), washed thoroughly, and tested. Both the $R_{MgCl2}$ (Fig. 5a) and flux (Fig. 5b) of the QSPIP-TMC membrane maintain stability during the 400-h immersion in 200 ppm NaClO. With further increasing NaClO concentration to 800 ppm, the $R_{MgCl2}$ of QSPIP-TMC membrane maintains high (>91%), while its flux increases slightly (Supplementary Fig. 14). By contrast, $R_{MgCl2}$ of PEI-TMC and BAPP-TMC membranes decrease from 97% and 98% to 14% and 4%, respectively, indicating structure degradation of both membranes as a result of NaClO treatment. Figure 5c–e shows the ATR-FTIR spectra of these membranes, in which the ~1640 and ~1482 $cm^{-1}$ peaks are assigned to amide bonds (A) and benzene rings (B). With increasing the immersion time, the ratio of the two peaks (A/B) for the QSPIP-TMC membrane remains stable (0.17 vs. 0.14). When it comes to PEI-TMC and BAPP-TMC membranes, the characteristic amide bond peak (1640 $cm^{-1}$) gradually decreases with increasing NaClO immersion time, while the benzene peak (1482 $cm^{-1}$) is stable. After 400 h immersion, the peak ratio (A/B) for the PEI-TMC membrane decreases from 0.23 to 0.09, while that for the BAPP-TMC membrane decreases from 0.57 to 0.04. These results indicate that the amide bonds in the QSPIP-TMC

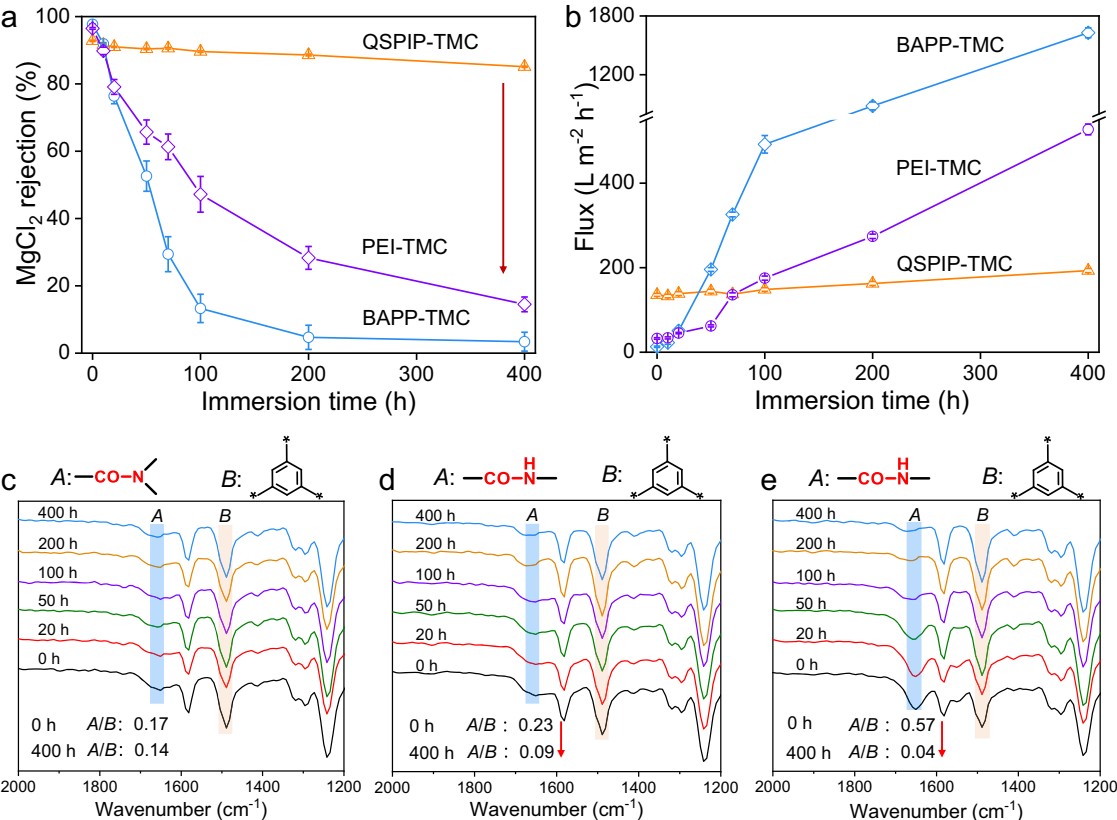

**Fig. 5 | Antichloride performance of membranes. a, b** Effect of immersion time (200 ppm NaClO) in MgCl₂ rejection and flux of QSPIP-TMC, PEI-TMC and BAPP-TMC membranes. ATR-FTIR spectra of **c** QSPIP-TMC, **d** PEI-TMC, and **e** BAPP-TMC membranes after being immersed in NaClO (200 ppm) at different times.

membrane are stable during the NaClO treatment, while those in PEI-TMC and BAPP-TMC membranes degrade significantly. The degradation of the amide bond is affected by active hydrogen atoms attached to the nitrogen in the amide bond.

As depicted in Fig. 6a, it is widely accepted that degradation of amide bonds under acidic conditions occurs via two key intermediates, i.e., *O*-chloroimidate I and *N*-chloroamide II[41–43]. As the formation of I via transition state III is the rate-determining step[42], we thus modeled the activation energy of III by density functional theory. Truncated blocks of QSPIP-TMC, PEI-TMC and BAPP-TMC were modeled at the M062X/6-311G** level of theory (Fig. 6b–d). Numerous attempts have been tried for the activation energy of QSPIP-TMC but failed to achieve convergence (Fig. 6b). The unconvergence is likely due to the strong intramolecular electrostatic repulsion of the two quaternary ammonium groups in QSPIP-TMC. On the other hand, the free activation energy is fairly low, being 8.01 and 6.70 kcal/mol for cyclic intermediates of PEI-TMC (Fig. 6c) and BAPP-TMC (Fig. 6d), respectively. These results suggest a facile occurrence of chlorination for PEI-TMC and BAPP-TMC at room temperature. Structurally, these results indicate that the poor chlorine resistance of control membranes is due to the sensitivity of the amidic nitrogen, and the QSPIP-TMC analog is stable.

Figure 6e–g shows that atomic content of chlorine element in QSPIP-TMC, PEI-TMC and BAPP-TMC membranes are 0.66, 2.77 and 5.27% after 400-h immersion in NaClO, while that in the pristine membranes are 0.30, 0.20 and 0.23%. That is, the chlorine contents in PEI-TMC and BAPP-TMC membranes were improved by ~14 and 23 times after NaClO treatment, consistent with the proposed mechanism that amide bonds in both membranes were attacked by chlorine *via* the active hydrogen. By contrast, the amide bond in QSPIP is chlorine resistant, and the changes in chlorine content in the QSPIP-TMC membrane are negligible. In addition, all the pristine QSPIP-TMC, PEI-TMC and BAPP-TMC membranes exhibit smooth, defect-free surfaces

prior to the NaClO immersion (Supplementary Fig. 15). After 400-h immersion in 200 ppm NaClO, surface morphology of the QSPIP-TMC maintain smooth and defect-free (Fig. 6h). By contrast, surfaces of both the PEI-TMC (Fig. 6i) and BAPP-TMC (Fig. 6j) membranes became rougher after 400-h NaClO immersion, with many pinholes (indicated by arrows) unequivocally seen on their surfaces. Meanwhile, the thickness of PEI-TMC and BAPP-TMC increased after the chlorine treatment, while the thickness of QSPIP-TMC membrane is stable (Supplementary Fig. 16). Collectively, both the chlorine content (Fig. 6e–g) and surface morphologies (Fig. 6h–j) of membranes agree with corresponding simulations (Fig. 6b–d), showing that the QSPIP structure is key to its chlorine resistance.

**Preparation of large-area membranes and modules**

Recently, McCutcheon et al. stressed the need for the scalability of novel membranes with advanced performances[58]. The QSPIP-TMC membranes were prepared by straightforward interfacial polymerization that is easy to operate. Taking this advantage, we conducted the QSPIP-TMC interfacial polymerization on a large-size PSf supporting membrane, leading to the formation of a large-area (1 × 2 m², Fig. 7a) membrane that is about two orders of magnitude larger than membranes commonly used in lab[26,28,58]. Five pieces of small-size membranes (M₁-M₅) were randomly selected from the large membrane (Supplementary Fig. 17), and they show morphologies (Supplementary Fig. 18) and separation performances (Fig. 7b) close to that of the small membrane (M₀). This indicates good quality control of the membrane preparation. Then the large-area QSPIP-TMC membrane was transformed into spiral wound modules (effective area: 0.5 m², Fig. 7c), which show reproducible R_MgCl2 and permeance that are stable in one-week nanofiltration and antichloride tests (Supplementary Fig. 19).

Then a streamlined separation protocol was designed to test the QSPIP-TMC module for proof-of-concept lithium extraction from high

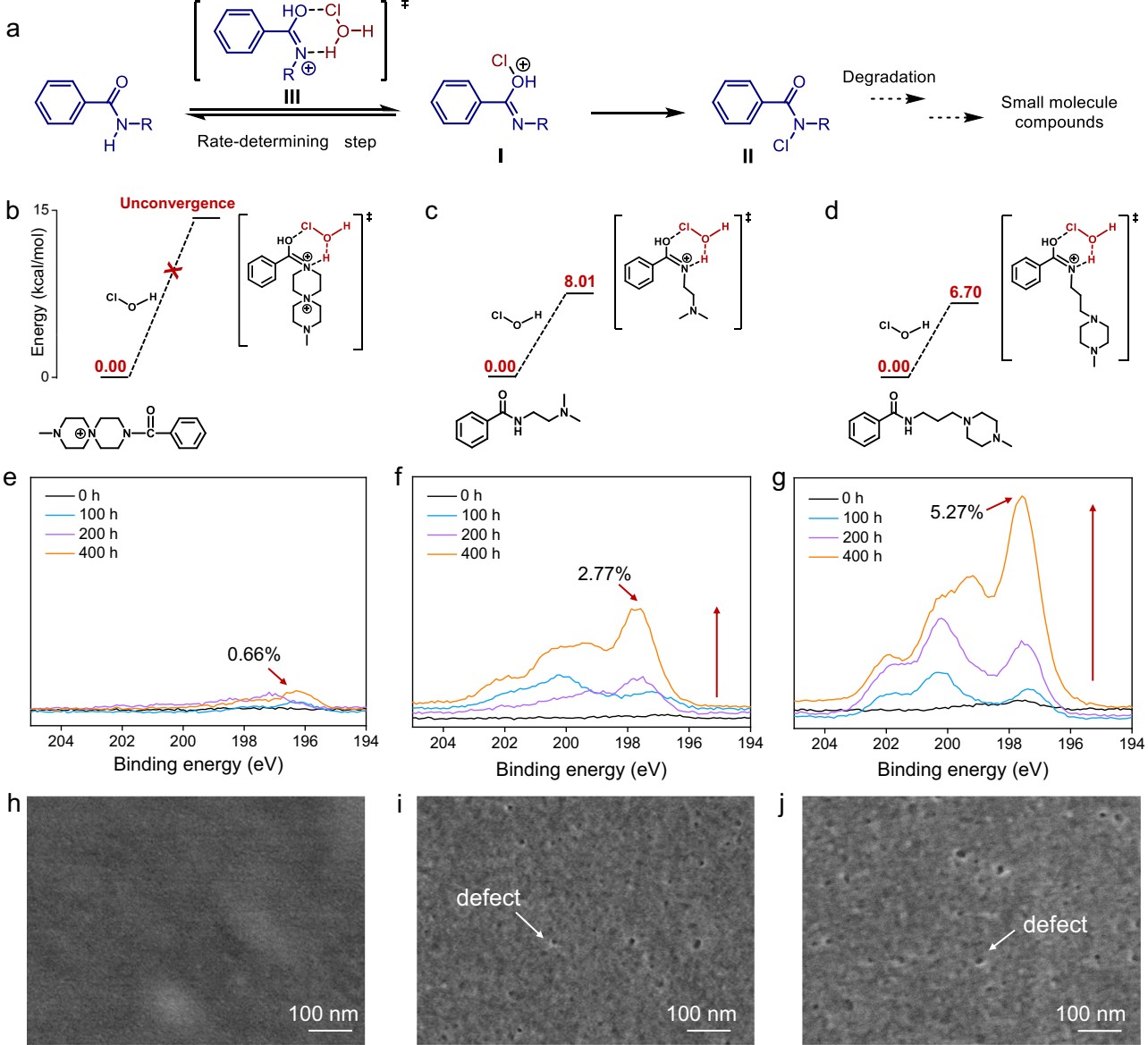

**Fig. 6 | Chlorine resistance mechanisms. a** Route of polyamide chlorination under acidic condition. Density functional theory calculation of transition state III of model molecules from **b** QSPIP-TMC, **c** PEI-TMC, **d** BAPP-TMC after NaClO treatment. XPS curves of **e** QSPIP-TMC, **f** PEI-TMC and **g** BAPP-TMC membranes after being immersed in NaClO (200 ppm) for different time. SEM surface morphologies of **h** QSPIP-TMC, **i** PEI-TMC and **j** BAPP-TMC membranes after 400-h NaClO immersion.

$Mg^{2+}/Li^+$ ratio brine. A simulated $Mg^{2+}/Li^+$ mixture (Fig. 7d) was filtered by three-stage nanofiltration with QSPIP-TMC modules (Fig. 7e). Figure 7f shows that ~20 L, ~33 L, ~34 L permeate solution were collected within 30 min by the 1st, 2nd and 3rd-stage nanofiltration, after which the $Mg^{2+}/Li^+$ ratio in the permeate was decreased to 7.56, 0.84, 0.05 (Fig. 7g), respectively. The permeate collected in the 3rd-stage nanofiltration was concentrated, and the $Li_2CO_3$ was precipitated by adding excessive $Na_2CO_3$ according to the literature method (Supplementary Fig. 20)[39,59]. It should be noted that this separation protocol (i.e., nanofiltration, precipitation) includes only the core steps for lithium extraction[59]. Nevertheless, the purity of $Li_2CO_3$ (97.2%, Fig. 7h) is readily high compared to literature results that also used three-stage nanofiltration treatment[39,60]. In practice, the purity of $Li_2CO_3$ could be improved by subsequent steps (e.g., recrystallization, carbonization, etc.) to prepare battery grade $Li_2CO_3$[61]. These results show that the QSPIP enables the preparation of high-permeance, chlorine-resistant, and scalable $Mg^{2+}/Li^+$ separation membranes.

In conclusion, a QSPIP monomer was designed to enable the straightforward preparation of chlorine-resistant, high-flux $Mg^{2+}/Li^+$ separation membranes by interfacial polymerization. The QSPIP features quaternary ammonia, spiral conformation and secondary amines, which simultaneously confer positive charge, higher fractional free volume, and chlorine resistance to QSPIP-TMC membranes. The water permeance of the QSPIP-TMC membrane is 5 times improved, which is attributed to the facilitated water transportation through the membrane. Meanwhile, the QSPIP monomer contains a secondary amine for condensation with acyl chloride groups from TMC, yielding a chlorine-resistant amide linkage that contains no active hydrogen. Thus the high separation performance of QSPIP-TMC membrane is stable versus 16 days of NaClO treatment. Large-area QSPIP-TMC membranes and spiral wound modules were prepared, both of which show reproducible and stable MgCl$_2$ rejection and permeance. Collectively, this work shows that the QSPIP represents a monomer that combines high $Mg^{2+}/Li^+$

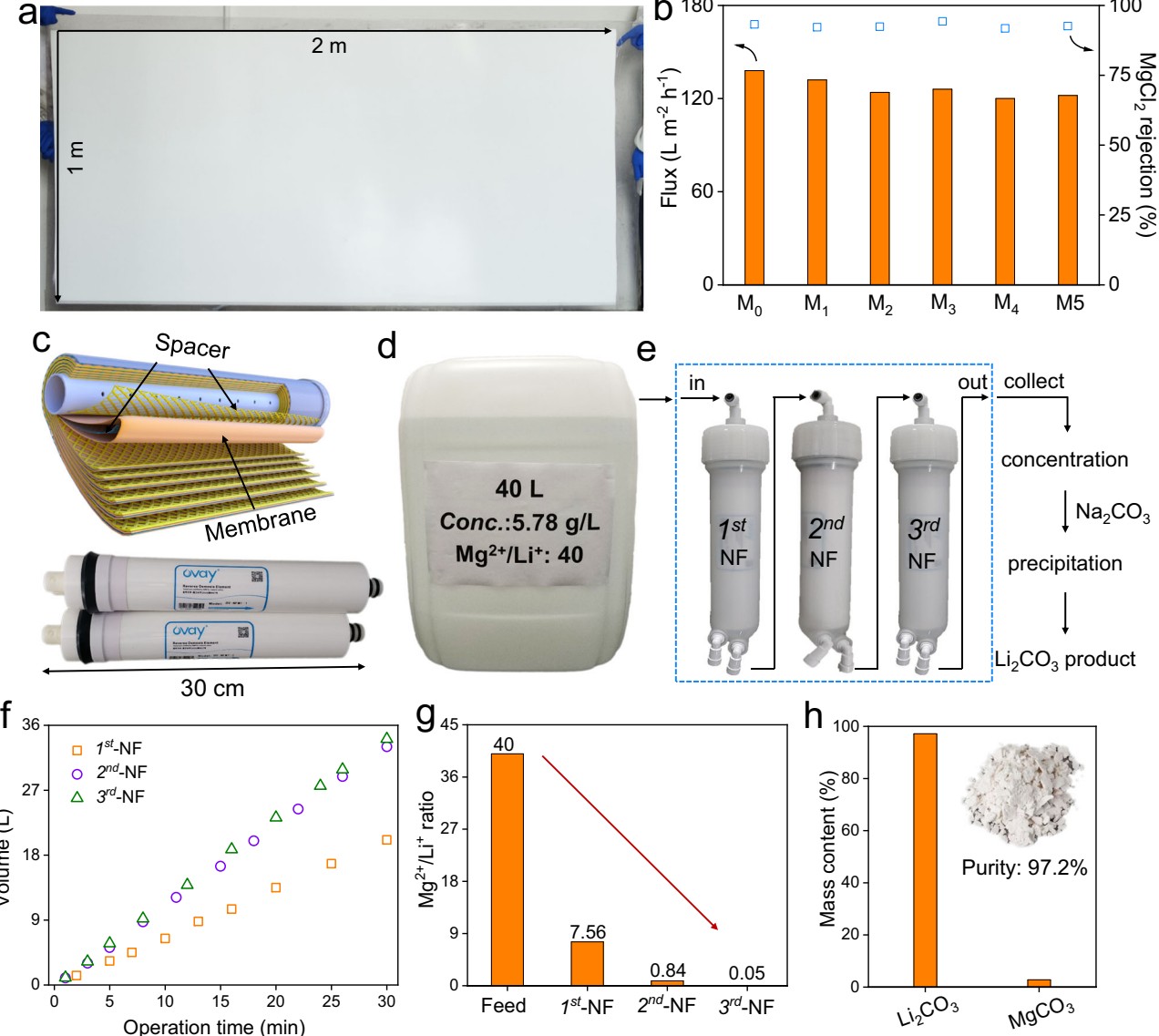

**Fig. 7 | Preparation and performance of large-area membranes and modules.**
**a** An optical image of a large-area QSPIP-TMC membrane, **b** nanofiltration performance of five pieces of small membranes ($M_1$-$M_5$, 8 × 8 cm$^2$) randomly selected from the membrane in (**a**). $M_0$ is the small membrane in Fig. 3b. Test conditions: 1000 ppm $MgCl_2$, 6 bar. **c** Schematic structures (top) and photographs (bottom) of QSPIP-TMC modules prepared from the membrane in (**a**). The optical image of **d** feed solution (salt concentration: 5.78 g/L, $Mg^{2+}/Li^+$ ratio: 40) and **e** three-stage nanofiltration. Please note: the feed solution was prepared to simulate the salt concentration and $Mg^{2+}/Li^+$ ratio in the East Taijinar Brine[61]. **f** Effect of operation time in the volume of permeates of 1st, 2nd, 3rd-stage nanofiltration. **g** The $Mg^{2+}/Li^+$ ratio of feed and permeates in 1st, 2nd, 3rd-stage nanofiltration. **h** The mass content of $Li_2CO_3$ and $MgCO_3$ in the $Li_2CO_3$ compound. The insert is the optical image of $Li_2CO_3$ compound.

separation performance, chlorine resistance and scalability in one single membrane.

## Methods

### Chemicals and materials

Piperazine (PIP, 99%), bis(2-chloroethyl)amine hydrochloride (98%), 1,3,5-trimesoyl chloride (TMC, 98%), 1,4-Bis(3-aminopropyl)piperazine (BAPP, 98%), Polyethyleneimine (PEI, 50 wt% in water, Mw: 70,000 Da), polyethylene glycol (Mw: 200, 400, 600, 1000, 1500 Da) were purchased from Shanghai Aladdin Reagent Company. Ethanol, n-hexane, and all inorganic salts were purchased from Sinopharm Chemical Reagents Co., Ltd. Polysulfone (PSf) membranes (-20 KDa) were bought from Separate Equipment Co., Ltd. (Beijing). Deionized water (DI, 18 MΩ) was produced by Water Purification System (Heal Force, China) and was used in all experiments.

### Synthesis of QSPIP

Piperazine (12 g, 0.140 mol) was dissolved in 60 mL DI water. Bis(2-chloroethyl)amine hydrochloride (27.4 g, 0.154 mol) was dissolved in 150 mL DI water and dropped into the piperazine solution. Then $CaCO_3$ (14.8 g, 0.148 mol) was added into the solution, and the mixture was refluxed at 55 °C for 9 h. After removing the white precipitates by filtration, the solution was concentrated by a rotary evaporator and precipitated in ethanol. The product was filtered, washed with ethanol three times, and dried in vacuum (50 °C, 8 h). The QSPIP yield: 62.8%. The chemical structure of QSPIP was confirmed by $^1$H/$^{13}$C-NMR (Fig. 1b, c), mass spectrum (Supplementary Fig. 1) and organic elemental analysis (Fig. 1d). $^1$H NMR (400 MHz, $D_2O$): δ = 3.99 (m, J = 5.5 Hz, 1H), 3.70 (m, 1H) ppm; $^{13}$C-NMR (101 MHz, $D_2O$): δ = 55.74, 37.25 ppm; HR-MS(ESI+): m/z: calculated for $C_8H_{18}N_3$: 156.1495, [M]$^+$ found: 156.1496.

## Preparation of QSPIP-TMC membrane

A piece of PSf membrane was fixed on a glass plate. QSPIP aqueous solution (0.5 wt%, pH = 11) was poured on the surface of PSf membrane and kept for 3 min. Then TMC hexane solution (0.1 wt%) was poured on the membrane surface, kept for 1 min, and finally heated in oven (50 °C, 10 min) to obtain the QSPIP-TMC membrane. Control membranes were prepared by interfacial polymerization of TMC with piperazine (PIP) polyethyleneimine (PEI), 1,4-Bis(3-aminopropyl) piperazine (BAPP) following the same method and named PIP-TMC, PEI-TMC, BAPP-TMC, respectively.

## Preparation of QSPIP/PIP-TMC polymer (for BET measurements in Fig. 2f)

TMC hexane solution (0.1 wt%, 400 mL) was mixed with QSPIP (or PIP) aqueous solution (0.5 wt%, 400 mL, pH = 11), and the mixture was vigorously stirred for 5 min. The precipitation was filtered and washed with hexane for 3 times, water for 3 times, ethanol for 3 times and water for 3 times. The product was freeze-dried and stored for BET measurement.

## Characterizations

Nuclear magnetic resonance (NMR) was conducted by Bruker AV400 using deuterium oxide as solvent. Mass spectroscopy (MS) was conducted on micrOTOF II (Bruker Switzerland) with electro spray ionization as the ionization source. The atomic content of nanofilm was determined by an Organic elemental analyzer (EA, Elemental Vario EL). The sample was heated in a vacuum oven (60 °C, 12 h) before test. The ATR-FTIR spectra were obtained by Thermo Scientific Nicolet IS5 spectrometer with a scan time of 16 and resolution rate of 16 cm$^{-1}$. X-ray Photoelectron Spectroscopy (XPS) was recorded by Thermo ESCALAB 250XI with a monochromatized Al Kα X-ray source. The carbon dioxide adsorption/desorption was measured by Micromeritics ASAP 2460 surface area analyzer at 0 °C. Samples were degassed at 120 °C for 12 h before test. The roughness of membranes was measured by a Scanning probe microscope (SPM, 9700 SPM, Shimadzu Japan). The surface morphologies of membranes were observed by Field emission scanning electron microscope (FESEM, SU8010, Japan). Samples were glued on the plate and sprayed with gold for 5 s before test. The voltage and current of the instrument were set at 3 kV and 10 μA. Transmission electron microscope (TEM) was obtained by Tecnai G2. The water contact angles were measured by contact angle measurement (OCA20, dataphysics). Zeta potentials of membranes were tested by streaming potential measurement (SurPASS3 electric analyzer, Anton Paar, Austria) with 1 mM KCl solution.

## Separation performance

Separation performance was evaluated by cross-flow (rate: 0.5 L min$^{-1}$) nanofiltration. Temperature of all feed was maintained 30 °C, and membranes were stabilized at 6 bar for 30 min before test. The flux (J), salt rejection (R) and Mg$^{2+}$/Li$^+$ selectivity (S) of membranes were calculated by the following equations:

$$J = \frac{V}{A \times t} \tag{1}$$

$$R = \frac{C_f - C_p}{C_f} \times 100\% \tag{2}$$

$$S = \frac{C_{f,Mg^{2+}}/C_{f,Li^+}}{C_{p,Mg^{2+}}/C_{p,Li^+}} \tag{3}$$

Above, V is volume of the permeate collected in given time t. A is the effective membrane area (A = 19.6 cm$^2$). $C_f$ and $C_p$ are the ion concentrations in the feed and permeate. The ion concentrations in SINGLE solution were confirmed by conductivity meter (DDS-307A,

China). In addition, ion concentration in MIXTURE solution was characterized by an inductively coupled plasma emission spectrometer (ICP-OES, iCAP 7000, Germany).

## Reporting summary

Further information on research design is available in the Nature Portfolio Reporting Summary linked to this article.

## Data availability

All the data is available from the corresponding author upon request. The coordinates of the optimized structures obtained by density functional theory calculation are provided in the Source Data file. Source data are provided with this paper.

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

## Acknowledgements

Q.Z. is grateful for the financial support from the National Key R&D Program of China (2022YFB3805103), National Natural Science Foundation of China (22178139), Key R&D Program of Hubei Province (2022BCA079), the Innovation and Talent Recruitment Base of New Energy Chemistry and Device (B21003), and Hubei Three Gorges Laboratory (No. SC212001). The authors thank Hunan Ovay Technology for the help with module preparation.

## Author contributions

Q.Z. conceived and supervised the project. H.P. performed the experimental studies with help from X.L. and J.L. The energy calculation and simulations were carried out by K.Y. and X.H. All authors wrote and revised the work.

## Competing interests

The authors declare no competing interests.
