## [Peer Review File · Nature Communications]

Quaternization-spiro design of chlorine-resistant and high permeance lithium separation membranesREVIEWER COMMENTS

Reviewer #1 (Remarks to the Author):

This work reported a novel monomer that improved both the chlorine resistance and permeance of Mg²⁺/Li⁺ separation membranes. The monomer consists of spiral quaternary ammonia, conferring chlorine resistant bonding and more free volume to QSPIP membranes. The structural and chemical properties of the prepared membranes are thoroughly characterized, and the results are backed up by characterization technologies and molecular simulation. Therefore, I am in favor of its publication after addressing the following comments:

1. Main novelty of this work is the design of QSPIP monomers. However, authors did not discuss thoroughly why they design QSPIP to combine the chlorine resistance with high permeance. More detailed discussion about designing principles of QSPIP should be provided.
2. In the Fig. 1, the FT-IR and XPS characterizations were done with freestanding QSPIP membranes, while the separation performance is tested with composite membranes. Authors should add such characterizations of QSPIP composite membranes.
3. In the Fig. 2b, authors used TEM to measure the thickness of QSPIP composite membranes. This characterization is appropriate, but how to define the boundary between the QSPIP selective layer and the PSf support from the Fig. 2b?
4. Interfacial polymerization is affected not only by the chemical structures of QSPIP, but also the reaction conditions. Since the monomer is new, it is necessary to know the modulation of polymerization conditions. What is the effect of reaction conditions, such as QSPIP/TMC concentrations, in separation performance?
5. In the Fig. 3e, authors showed the rejection of QSPIP membrane to different ions, and the membrane permeance are needed as well. Moreover, the corresponding references of the data for comparison in Fig. 3g should be specified in the supplementary information.

6. In the Fig. 5, the chlorine resistance of the PEI and BAPP control membranes are poor, and the surface morphologies of both membranes indicate their degradation. What is the effect of chlorine treatment in thickness of these membranes?

Reviewer #2 (Remarks to the Author):

The paper describes interesting new chemistry for fabricating nanofiltration membranes via interfacial polymerization. The membranes have an excellent performance in the selective separation Li^+ of high ratio $\text{Mg}^{2+}/\text{Li}^+$ brines and are very resistant to degradation by chlorine.

The study is comprehensive and original, and the quality and methodology of the work are sound. The paper is certainly interesting for researchers in the field of membrane science and is suitable for high-ranking specific Journals for that field.

But, in my view, the work is less suitable for the broader audience of Nature Communications. The paper describes in detail the membranes and their properties, but not in the actual context of advances in the targeted application. It is unclear if the high chlorine resistance indeed solves a show-stopper of membrane technology for Li^+ recovery, and how such technology compares to the current state of the art. For publication Nature Communications, the paper should provide a proof-beyond-doubt that, as compared to existing technologies and other currently developed new technologies, the new membranes indeed provide a great step forwards in recovery of Lithium.

Thus, despite the high quality and originality of the work, I recommend that the authors submit this work to another high-ranking more specialized Journal

Reviewer #3 (Remarks to the Author):

On the whole, Peng et al. designed a quaternized-spiral piperazine (QSPIP) monomer featuring spiral conformation and quaternary/secondary amines, significantly elevating the permeance and chlorine-resistance of nanofiltration membrane for lithium extraction. The

permeance of QSPIP-TMC membrane is $22 \text{ L m}^{-2} \text{ h}^{-1} \text{ bar}^{-1}$, and the perm-selectivity is stable after 400-h immersion against NaClO. And molecular simulation results compared the fractional free volume and chlorination energy between QSPIP-TMC and other common polyamides to evidence the underlying mechanism. This research focuses on the practical issues and proposes efficient solution with informative data and well-written words. The reviewer evaluates that the current version of the manuscript needs further modification according to the following comments:

1. The QSPIP yield for the cyclization reaction between piperazine (PIP) and bis(2-chloroethyl)amine should be provided in the experiment. The mass spectroscopy could measure the molecular weight of QSPIP.
2. At page 6, line 111, "It is noted that QSPIP is one of the rare strong electrolyte exploited as standalone monomers for straightforward preparation of nanofiltration membranes". It is suggested to describe the advantages of QSPIP compared to other electrolytes that free QSPIP from the permanent limitations.
3. At page 7, line 139, "The orange and gray colors indicate section voids and inner voids, respectively". The gray color indicates the occupied space of polymer skeleton that might not belong to free volume.
4. In page 6, line 120-125, it demonstrates that the spiral configuration of QSPIP increases the free volume of the polymer and thus enhances the microporosity. It is suggested to further compare QSPIP-TMC with BAPP-TMC, since BAPP has similar sizes but different molecular configuration?
5. Fig. 4 presents the difference in the angles of water passage through the QSPIP and PIP, where the water has two transport angles for QSPIP because of the nonplanar conformation and strong interaction of quaternary ammonium and water. This result is interesting, but it is commonly considered that the transport channels is the voids between the polymer skeleton. Whether the authors propose that the rings of piperazine can also act as transport channels?
6. This work has proved the high stability of QSPIP-TMC in 200 ppm NaClO solution. Could the QSPIP-TMC endure higher NaClO concentration?
7. Please supply the samples preparation method for BET measurement in method section.

Point to point response

Reviewer #1: This work reported a novel monomer that improved both the chlorine resistance and permeance of Mg^{2+}/Li^+ separation membranes. The monomer consists of spiral quaternary ammonia, conferring chlorine resistant bonding and more free volume to QSPIP membranes. The structural and chemical properties of the prepared membranes are thoroughly characterized, and the results are backed up by characterization technologies and molecular simulation. Therefore, I am in favor of its publication after addressing the following comments:

1. Main novelty of this work is the design of QSPIP monomers. However, authors did not discuss thoroughly why they design QSPIP to combine the chlorine resistance with high permeance. More detailed discussion about designing principles of QSPIP should be provided.

Response: Thanks a lot. Currently, PEI-based membranes are being widely studied for Mg^{2+}/Li^+ separation. These membranes show limited permeance, and moreover, the amide groups (*e.g.*, -CO-NH-) in them are susceptible to NaClO. To these ends, we are motivated to: 1) reduce the network crosslinking of membranes, 2) design monomers with non-planar conformation to enable loosen structures of membranes, 3) eliminate chlorine-sensitive amide structures by using secondary amines (-NH) as monomers. Chemical structure of QSPIP fullfills these requirments and the QSPIP-TMC membrane shows both high permeance and chlorine resistance.

Revisions made:

Page 3, Line 14: To this end, reacting monomers should contain secondary amines (-NH) rather than primary amines (-NH₂) to eliminate chlorine-sensitive amide bonds in resultant membranes. Meanwhile, positive charges are needed for Mg^{2+}/Li^+ separation, while the conformation of monomers could be contorted to increase the microporosity of resultant membranes.

2. In the Fig. 1, the FT-IR and XPS characterizations were done with freestanding QSPIP membranes, while the separation performance is tested with composite membranes. Authors should add such characterizations of QSPIP composite membranes.

Response: Thanks a lot. Both the ATR-FTIR and XPS characterizations of QSPIP-TMC composite membranes were added (Fig. R1). The composite membrane shows signals of both the amine (3460 cm^{-1} , from QSPIP) and amide group (1640 cm^{-1} , due to the QSPIP-TMC condensation) in ATR-FTIR spectra (Fig. R1a). Meanwhile, the N atomic content of QSPIP-TMC membrane is 10.4%, 4.7 times higher than PSf membrane (Fig. R1b). The XPS peak at 401.7 eV corresponding to quaternary ammonium group is seen on QSPIP-TMC membrane (Fig. R1c). These results are consistent with the QSPIP-TMC freestanding membrane (Fig. 1, main text).

Fig. R1. (a) The ATR-FTIR, (b) XPS characterizations of PSf and QSPIP-TMC composite membrane. (c) The N 1s core level spectra of QSPIP-TMC composite membrane.

Revisions made: Fig. R1 was added as Supplementary Fig. 5.

Page 5, Line 9: Both the ATR-FTIR and XPS characterizations confirmed that QSPIP-TMC membranes have been prepared on the PSf supports (Supplementary Fig. 5).

Supplementary Information: Page 7, Line 3: Discussions: The QSPIP-TMC composite membrane shows signals of both the amine (3460 cm^{-1} , from QSPIP) and amide group (1640 cm^{-1} , due to the QSPIP-TMC condensation) in ATR-FTIR spectra. The N atomic content of QSPIP-TMC membrane is 10.4%, 4.7 times higher than PSf membrane. The XPS peak at 401.7 eV corresponding to quaternary ammonium group is seen on QSPIP-TMC membrane. These results are consistent with the QSPIP-TMC freestanding membrane (Fig. 1, main text).

3. In the Fig. 2b, authors used TEM to measure the thickness of QSPIP composite membranes. This characterization is appropriate, but how to define the boundary between the QSPIP selective layer and the PSf support from the Fig. 2b?

Response: Thanks a lot. The boundary of QSPIP-TMC layer and the PSf support was determined by the brightness contrast. In detail, the darker and brighter regions are assigned to QSPIP-TMC layer and PSf layer, respectively. This is likely because the density of QSPIP-TMC layer is higher than the PSf layer and epoxy resin, and it is more difficult for electron beam to pass through the QSPIP-TMC layer^{R1}. As an evidence, a dark region appears between the resin and PSf layer in the QSPIP-TMC membrane, which is not observed in the PSf supporting membrane (Fig. R2), confirming that the dark region in QSPIP-TMC membrane is QSPIP-TMC layer.

Reference:

[R1] Yamasaki, J., Ubata Y. & Yasuda H. Empirical determination of transmission attenuation curves in mass-thickness contrast TEM imaging. *Ultramicroscopy* **200**, 20-27 (2019).

Fig. R2. The TEM morphologies of (a) QSPIP-TMC, (b) PSf membrane.

Revisions made:

Page 7, Line 3: Note: the dark region is QSPIP-TMC due to the density contrast.

4. Interfacial polymerization is affected not only by the chemical structures of QSPIP, but also the reaction conditions. Since the monomer is new, it is necessary to know the modulation of polymerization conditions. What is the effect of reaction conditions, such as QSPIP/TMC concentrations, in separation performance?

Response: Thanks a lot. The suggested experiments were added in Fig. R3. With increasing QSPIP concentration from 0.1 wt% to 2.0 wt%, the flux of QSPIP-TMC first increased and then decreased. When the QSPIP concentration is 0.5 wt%, the flux and $MgCl_2$ rejection (R_{MgCl_2}) of QSPIP-TMC were $\sim 138 \text{ L m}^{-2} \text{ h}^{-1}$ and 92.5%. Meanwhile, with increasing TMC concentration from 0.05 wt% to 0.5 wt%, the flux of QSPIP-TMC decreased from 152.6 to $53.7 \text{ L m}^{-2} \text{ h}^{-1}$. With increasing reaction time of QSPIP/TMC to 120 s, the flux of QSPIP-TMC decreased to $62.7 \text{ L m}^{-2} \text{ h}^{-1}$, while R_{MgCl_2} increased to 94.1%. On basis of Fig. R3, concentration of QSPIP and TMC were kept at 0.5 wt% and 0.1 wt%, and the reaction time of QSPIP/TMC was 60 s.

Fig. R3. (a) Effect of QSPIP concentration in separation performance of QSPIP-TMC membrane. (TMC: 0.1 wt%, condensation time: 60 s). (b) Effect of TMC concentration in separation performance of QSPIP-TMC membrane. (QSPIP: 0.5 wt%, condensation time: 60 s). (c) Effect of condensation time of QSPIP/TMC in separation performance of QSPIP-TMC membrane. (QSPIP: 0.5 wt%, TMC: 0.1 wt%). Test conditions: 1000 ppm $MgCl_2$, 6 bar, 30°C.

Revisions made: Fig. R3 was added as Supplementary Fig. 4.

Page 5, Line 7: Here, the interfacial polymerization conditions were tuned to optimize the separation performance of QSPIP-TMC membrane (Supplementary Fig. 4).

Supplementary Information: Page 6, Line 17: After optimization of preparation conditions of QSPIP-TMC, concentration of QSPIP and TMC were kept at 0.5 wt% and 0.1 wt%, and the condensation time of QSPIP/TMC was 60 s.

5. In the Fig. 3e, authors showed the rejection of QSPIP membrane to different ions, and the membrane permeance are needed as well. Moreover, the corresponding references of the data for comparison in Fig. 3g should be specified in the supplementary information.

Response: Thanks a lot. The flux of QSPIP-TMC membrane towards different ions were added in Fig. R4, which are $\sim 140 \text{ L m}^{-2} \text{ h}^{-1}$ at 6 bar. In addition, the corresponding references of the data in Fig. 3g were specified in the Table R1. Please see the revisions below.

Fig. R4. The flux of QSPIP-TMC membrane towards different ions. (Conditions: 1000 ppm, 6 bar).

Table R1. Original data and references for performance comparison in Fig. 3g

Membrane	C _{feed} (ppm)	Mg ²⁺ /Li ⁺ ratio in feed	Permeance (LMH/bar)	Mg ²⁺ /Li ⁺ selectivity	Ref
PIP-MWCNTs	2000	21.4	14	7.1	14
PEI-TMC	2000	20	5.2	20	15
BPEI/TMC/EDTA	2500	24	0.6	9.2	16
PEI/TMC/CNC-COOH	2000	30	4.2	12.2	17
PEI/TMC/CNC-COOH	2000	60	3.4	5.8	17
DAPP-TMC	2000	20	2.5	2.6	18
PHF-doped TFC	2000	21.4	6.3	13.1	19
[MimAP][Tf ₂ N]-PA	2000	20	4.7	8.1	20
(PES-GO)/PEI/TMC	2000	20	11.2	16.1	21
Dual-skin layer NF	2000	21.4	12	33.4	22
MBCN-0.02	2000	73	5.6	23.9	23
PEI/GQDs-NH ₂ /TMC	2000	20	11.9	27.8	24
PES/CQDs-NH ₂ /TMC	2000	20	12	14.4	25
Cu-MPD membrane	2000	23	16.3	8	26
PEI-TMC-QBPD	2000	50	13.6	5.9	27

PEI-TMC-HMTAB	2000	50	16.3	10.2	28
PEI@15C5	2000	20	8	14	29
PIL-TMC	2000	100	10	9.8	30
PBI_12-25K	2000	10	2.8	15.2	31
PEI-LDH/GA/PAN	1000	10	4	18.7	32
IP membrane	2000	20	0.4	3.3	33
SERS-0.50	2000	20	2.3	7.7	33
SIP-0.15	2000	20	1.3	15.4	33
(MWCNTs-COOK)-PEI	2000	20	12.2	58	34
PEI-TMC-QEDTP	2000	50	18.8	15.6	35
QSPIP-TMC	2000	100	22.2	9.1	This work

Revisions made: Fig. R4 was added as Supplementary Fig. 12. Table R1 was added as Table S1.

Page 8, Line 20: The flux of QSPIP-TMC towards feeds containing these mono/divalent ions are $\sim 140 \text{ L m}^{-2} \text{ h}^{-1}$ (Supplementary Fig. 12).

Page 9, Line 14: Note: the corresponding references in (g) were specified in the Table S1.

6. In the Fig. 5, the chlorine resistance of the PEI and BAPP control membranes are poor, and the surface morphologies of both membranes indicate their degradation. What is the effect of chlorine treatment in thickness of these membranes?

Response: Thanks a lot. As shown by the added SEM (Fig. R5), the thickness of QSPIP-TMC membrane was stable ($\Delta \approx 4 \text{ nm}$) after 400-h chlorine treatment. By contrast, the thickness of PEI-TMC and BAPP-TMC membranes increased more obviously ($\Delta_{\text{PEI-TMC}} \approx 16 \text{ nm}$, $\Delta_{\text{BAPP-TMC}} \approx 18 \text{ nm}$). This is because the polyamide layer of PEI-TMC and BAPP-TMC membranes are susceptible to chlorine-degradation during the chlorine treatment, leading to the looser selective layer of membranes. Such observations are consistent with previous reports^{R1, R2}.

Reference:

[R1] Hashiba, K., Nakai S., Ohno M., Nishijima W., Gotoh T. & Iizawa T. Deterioration mechanism of a tertiary polyamide reverse osmosis membrane by hypochlorite. *Environ. Sci. Technol.* **53**, 9109-9117 (2019).

[R2] Verbeke, R., Gómez V. & Vankelecom I. F. J. Chlorine-resistance of reverse osmosis (RO) polyamide membranes. *Prog. Polym. Sci.* **72**, 1-15 (2017).

Fig. R5. The thickness of (a, d) QSPIP-TMC, (b, e) PEI-TMC, (c, f) BAPP-TMC membranes before (a-c) and after (d-f) chlorine treatment for 400 h.

Revisions made: Fig. R5 was added as Supplementary Fig. 16.

Page 14, Line 1: Meanwhile, the thickness of PEI-TMC and BAPP-TMC increased after the chlorine treatment, while the membrane thickness of QSPIP-TMC membrane is stable (Supplementary Fig. 16).

Reviewer #2: The paper describes interesting new chemistry for fabricating nanofiltration membranes via interfacial polymerization. The membranes have an excellent performance in the selective separation Li^+ of high ratio $\text{Mg}^{2+}/\text{Li}^+$ brines and are very resistant to degradation by chlorine. The study is comprehensive and original, and the quality and methodology of the work are sound. The paper is certainly interesting for researchers in the field of membrane science and is suitable for high-ranking specific Journals for that field. But, in my view, the work is less suitable for the broader audience of Nature Communications. The paper describes in detail the membranes and their properties, but not in the actual context of advances in the targeted application. It is unclear if the high chlorine resistance indeed solves a show-stopper of membrane technology for Li^+ recovery, and how such technology compares to the current state of the art. For publication Nature Communications, the paper should provide a proof-beyond-doubt that, as compared to existing technologies and other currently developed new technologies, the new membranes indeed provide a great step forwards in recovery of Lithium. Thus, despite the high quality and originality of the work, I recommend that the authors submit this work to another high-ranking more specialized Journal.

Response: Thanks a lot. We appreciate the reviewer approving the high quality and originality of our work. Meanwhile the reviewer raised high standard on the broad audience and technology advances of the work. In addition to addressing all concerns from reviewer#1 and #3, we added new data according to reviewer#2's suggestions, and showed the concrete progress made in this work and why it will appeal to broad audience.

1) Broad audience. We are pleased to notice that the Nature Communications has continuous interests in novel membranes addressing energy/environmental challenges, and rising numbers of high-quality membrane papers were published. Among them, polyamide composite membranes are one of the focal points of current research (Representative refs in 2023: *Nat. Commun.*, 2023, 14, 1112. *Nat. Commun.*, 2023, 14, 2373. *etc.*), because they are one of the most widely used membranes in practical nanofiltration and desalination. Our work designed a new monomer which simultaneously improved the $\text{Mg}^{2+}/\text{Li}^+$ separation, chlorine resistance, and materials scalability (supported by new data added in this revision). Such membranes have not been reported. While these new results appeal to the membrane community that is readily large, it will also appeal to related fields of polymer chemistry, materials, and lithium-related new energy.

2) Preparation of large-area membranes and modules. Very recently, McCutcheon *et al.*, in their latest *Science* paper stressed the urgent need for **novel-and-scalable** membranes^{R1}. Many new membranes are emerging to enhance $\text{Mg}^{2+}/\text{Li}^+$ separation performance, but most of them are small-sized due to the preparation complexity, and $\text{Mg}^{2+}/\text{Li}^+$ modules made from these novel membranes were not reported^{R2-R4}. The QSPIP monomer enables the straightforward interfacial polymerization, which is facile and renders the scalable preparation of membranes with good quality-control. As shown by the new data (Fig. R6a), we succeeded to prepare a large-area QSPIP-TMC membrane ($1.0 \times 2.0 \text{ m}^2$), and transformed it into spiral wound modules (effective area: 0.5 m^2 , Fig. R6b). The morphologies of large-area QSPIP-TMC membrane are consistent with that of the small-sized membrane (Fig. R7). Meanwhile, both the flux and R_{MgCl_2} of large-area QSPIP-TMC membrane ($\sim 22 \text{ L m}^{-2} \text{ h}^{-1} \text{ bar}^{-1}$, R_{MgCl_2} : $\sim 92\%$, Fig. R6c) and modules ($\sim 23 \text{ L m}^{-2} \text{ h}^{-1} \text{ bar}^{-1}$, R_{MgCl_2} : $\sim 90.2\%$, Fig. R6d) are on par with that of the small membrane (M_0). In addition to the good performance, the scalability of QSPIP-TMC is a beneficial step forward.

Fig. R6. (a) Optical image of a piece of large-area QSPIP-TMC membrane ($1.0 \times 2.0 \text{ m}^2$). (b) optical images (left) and schematic structures (right) of spiral wound modules made from the membrane in (a). (c) Separation performance of five pieces of small-size membranes (M_1 - M_5 ,

$8.0 \times 8.0 \text{ cm}^2$) randomly selected from the membrane in (a). Note: M_0 was the small membrane in Fig. 3a. (d) Separation performance of the QSPIP-TMC module during 7-day nanofiltration test. (Conditions: 1000 ppm MgCl_2 , 6 bar).

Fig. R7. Surface morphologies of (a) M_0 and (b-f) M_1 - M_5 membranes.

3) Advanced overall separation performance compared to current technology. Currently, the “adsorption-membrane separation” method represents cutting-edge technology for lithium extraction from high $\text{Mg}^{2+}/\text{Li}^+$ ratio brines^{R5, R6}. This technology is green and economic compared to solvent extraction and calcination. In this regard, nanofiltration membranes play crucial roles to reduce the $\text{Mg}^{2+}/\text{Li}^+$ ratio. Compared with commercial membranes such as DK, DL, NF270, the QSPIP-TMC shows 2~5 times improved permeances, while the $\text{Mg}^{2+}/\text{Li}^+$ selectivity of QSPIP-TMC is close to them^{R7-R9}. In addition, membranes are frequently fouled during the practical recovery process, and NaClO was widely used to reduce biological growth and clean the membranes^{R10}. Both the QSPIP-TMC membranes and modules feature good chlorine resistance (Fig. R8), which is beneficial for practical application. Collectively, the QSPIP-TMC membrane shows advances in terms of the overall separation performance compared to current technologies.

Fig. R8. Separation performance of large-area QSPIP-TMC membranes and modules before and after being treated by 200 ppm NaClO for 48 h. Note: M_6 - M_8 was randomly selected from the large-area QSPIP-TMC membrane in Fig. R6a. (Test conditions: 1000 ppm MgCl_2 , 6 bar).

4) Practical utility of the QSPIP-TMC modules. Considering the compositions of representative high Mg^{2+}/Li^+ ratio brins in China (e.g., East Taijinar Brine), we prepared a simulated brine (total concentration: 5.78 g/L, Mg^{2+}/Li^+ ratio: 40)^{R5}, and filtered it by three-stage nanofiltration (Fig. R9a). Fig. R9b shows that ~20 L, ~33 L, ~34 L feed solution were filtered within 30 min by the 1st, 2nd and 3rd-stage nanofiltration, after which the Mg^{2+}/Li^+ ratio was reduced to 7.56, 0.84, 0.05 (Fig. R9c), respectively. Then the permeate collected in 3rd-stage NF was concentrated and the Li_2CO_3 was precipitated *via* Na_2CO_3 precipitation according to literature method^{R11}.

Please note: the separation protocol in Fig. R9a includes only the core separation steps of a lithium extraction technology. That is, a practical lithium extraction line contains more separation steps^{R12}. Despite the streamlined protocols in Fig. R9a, the purity of Li_2CO_3 (97.2%, Fig. R9d) is readily high, and could be improved by industrial method (e.g., recrystallization, carbonization) to prepare battery-standard Li_2CO_3 ^{R5}. The purity of Li_2CO_3 produced in Fig. R9a is higher than Li_2CO_3 prepared by similar protocols in literature (e.g., 93%, 95%)^{R11, R13}. This proof-of-concept experiment shows the practical potential of QSPIP-TMC in lithium extraction.

Fig. R9. (a) Schematic producing of Li_2CO_3 by three-stage nanofiltration, which includes feed tank (40 L, concentration: 5.78 g/L, Mg^{2+}/Li^+ ratio: 40), nanofiltration units, concentration of permeate, precipitation and Li_2CO_3 . The permeate of 1st, 2nd-stage nanofiltration were used as the feed solution for the 2nd and 3rd-stage nanofiltration. (b) Effect of operation time in the volume of permeates of 1st, 2nd, 3rd-stage nanofiltration. (c) The Mg^{2+}/Li^+ ratio of feed and permeates in 1st, 2nd, 3rd-stage nanofiltration. (d) The mass content of Li_2CO_3 and $MgCO_3$ in the product.

We appreciated that your suggestions helped us to improve the comprehensiveness and significance of our work. The revised work not only prepared high-permeance and chlorine-resistant Mg^{2+}/Li^+ separation membranes, but also makes a concrete step forward in preparing scalable membranes and modules. We believe the revised work could appeal to broader audience.

Reference:

- [R1]. McCutcheon, J. R. & Mauter M. S. Fixing the desalination membrane pipeline. *Science* **380**, 242-244 (2023).
- [R2]. Vera, M. L., Torres W. R., Galli C. I., Chagnes A. & Flexer V. Environmental impact of direct lithium extraction from brines. *Nat. Rev. Earth Environ.* **4**, 149-165 (2023).
- [R3]. Razmjou, A., Asadnia M., Hosseini E., Habibnejad Korayem A. & Chen V. Design principles of ion selective nanostructured membranes for the extraction of lithium ions. *Nat. Commun.* **10**, 5793 (2019).
- [R4]. Hou, J., Zhang H., Thornton A. W., Hill A. J., Wang H. & Konstas K. Lithium extraction by emerging metal-organic framework-based membranes. *Adv. Funct. Mater.* **31**, 2105991 (2021).
- [R5]. Zhang, T., Zheng W. J., Wang Q. Y., Wu Z. C. & Wang Z. W. Designed strategies of nanofiltration technology for Mg^{2+}/Li^+ separation from salt-lake brine: A comprehensive review. *Desalination* **546**, 116205 (2023).
- [R6]. Liu, G., Zhao Z. & Ghahreman A. Novel approaches for lithium extraction from salt-lake brines: A review. *Hydrometallurgy* **187**, 81-100 (2019).
- [R7]. Li, Y., Zhao Y., Wang H. & Wang M. The application of nanofiltration membrane for recovering lithium from salt lake brine. *Desalination* **468**, 114081 (2019).
- [R8]. Ashraf, M. A., Li X. C., Wang J. F., Guo S. W. & Xu B. H. DiaNanofiltration-based process for effective separation of Li^+ from the high Mg^{2+}/Li^+ ratio aqueous solution. *Sep. Purf. Technol.* **247**, 116965 (2020).
- [R9]. Awais Ashraf, M., Usman M., Hussain I., Ahmad F., Guo S. & Zhang L. Lithium extraction from high magnesium salt lake brine with an integrated membrane technology. *Sep. Purf. Technol.* **302**, 122163 (2022).
- [R10]. Verbeke, R., Gómez V. & Vankelecom I. F. J. Chlorine-resistance of reverse osmosis (RO) polyamide membranes. *Prog. Polym. Sci.* **72**, 1-15 (2017).
- [R11]. Li, H. W., *et al.* Nanofiltration membrane with crown ether as exclusive Li^+ transport channels achieving efficient extraction of lithium from salt lake brine. *Chem. Eng. J.* **438**, 135658 (2022).
- [R12]. Haddad, A. Z., Hackl L., Akuzum B., Pohlman G., Magnan J. F. & Kostecki R. How to make lithium extraction cleaner, faster and cheaper — in six steps. *Nature* **616**, 245-248 (2023).
- [R13]. Li, T. Y., Zhang X. Z., Zhang Y., Wang J. X., Wang Z. & Zhao S. Nanofiltration membrane comprising structural regulator Cyclen for efficient Li^+/Mg^{2+} separation. *Desalination* **556**, 116575 (2023).

Revisions made: Fig. R6a-c, Fig. R9 were put together and added as Fig. 7 in the main text. Fig. R7 was added as Supplementary Fig.18. Fig. R6d, R8 was added as Supplementary Fig. 19.

Page 1, Line 18: translating its beneficial properties into large-area membranes ($1 \times 2 \text{ m}^2$).

Page 1, Line 22: which were reproducible at the membrane module level.

Page 15, Line 1: Preparation of large-area membranes and modules. Recently, McCutcheon *et al.*, stressed the need for the scalability of novel membranes with advanced performances⁵⁸.

The QSPIP-TMC membranes were prepared by straightforward interfacial polymerization that is easy-to-operate. Taking this advantage, we conducted the QSPIP-TMC interfacial polymerization on a large-size PSf supporting membrane, leading to the formation of a large-area ($1 \times 2 \text{ m}^2$, Fig. 7a) membrane that is about two orders of magnitude larger than membranes commonly used in lab^{26, 28, 58}. Five pieces of small-size membranes (M_1 - M_5) were randomly selected from the large membrane (Supplementary 17), and they show morphologies (Supplementary 18) and separation performances (Fig. 7b) close to that of the small membrane (M_0). This indicates good quality-control of the membrane preparation. Then the large-area QSPIP-TMC membrane was transformed into spiral wound modules (effective area: 0.5 m^2 , Fig. 7c), which show reproducible R_{MgCl_2} and permeance that are stable in one-week nanofiltration and antichloride tests (Supplementary Fig. 19).

Then a streamlined separation protocol was designed to test the QSPIP-TMC module for proof-of-concept lithium extraction from high $\text{Mg}^{2+}/\text{Li}^+$ ratio brine. A simulated $\text{Mg}^{2+}/\text{Li}^+$ mixture (Fig. 7d) was filtered by three-stage nanofiltration with QSPIP-TMC modules (Fig. 7e). Fig. 7f shows that $\sim 20 \text{ L}$, $\sim 33 \text{ L}$, $\sim 34 \text{ L}$ permeate solution were collected within 30 min by the 1st, 2nd and 3rd-stage nanofiltration, after which the $\text{Mg}^{2+}/\text{Li}^+$ ratio in the permeate was decreased to 7.56, 0.84, 0.05 (Fig. 7g), respectively. The permeate collected in the 3rd-stage nanofiltration was concentrated and the Li_2CO_3 was precipitated by adding excessive Na_2CO_3 according to literature method (Supplementary Fig. 20)^{39, 59}. It should be noted that this separation protocol (*i.e.*, nanofiltration, precipitation) includes only the core steps for lithium extraction⁵⁹. Nevertheless the purity of Li_2CO_3 (97.2%, Fig. 7h) is readily high compared to literature results that also used three-stage nanofiltration treatment^{39, 60}. In practical, the purity of Li_2CO_3 could be improved by subsequent steps (*e.g.*, recrystallization, carbonization, *etc.*) to prepare battery grade Li_2CO_3 ⁶¹. These results show that the QSPIP enables the preparation of high permeance, chlorine resistant, and scalable $\text{Mg}^{2+}/\text{Li}^+$ separation membranes.

Page 17, Line 12: Large-area QSPIP-TMC membranes and spiral wound modules were prepared, both of which show reproducible and stable MgCl_2 rejection and permeance.

Reviewer #3: On the whole, Peng et al. Designed a quaternized-spiral piperazine (QSPIP) monomer featuring spiral conformation and quaternary/secondary amines, significantly elevating the permeance and chlorine-resistance of nanofiltration membrane for lithium extraction. The permeance of QSPIP-TMC membrane is $22 \text{ L m}^{-2} \text{ h}^{-1} \text{ bar}^{-1}$, and the perm-selectivity is stable after 400-h immersion against NaClO . And molecular simulation results compared the fractional free volume and chlorination energy between QSPIP-TMC and other common polyamides to evidence the underlying mechanism. This research focuses on the practical issues and proposes efficient solution with informative data and well-written words. The reviewer evaluates that the current version of the manuscript needs further modification according to the following comments:

1. The QSPIP yield for the cyclization reaction between piperazine (PIP) and bis(2-chloroethyl)amine should be provided in the experiment. The mass spectroscopy could measure the molecular weight of QSPIP.

Response: Thanks a lot. The yield of QSPIP (62.8%) were added in the experimental part of QSPIP synthesis. The mass spectroscopy of QSPIP was added in Fig. R10. A peak at $m/z = 156.15$ was observed, which is consistent with the chemical structure of QSPIP ($C_8H_{18}N_3^+$, $M_w = 156$). The peak at 157.15 is formed by the interaction of H^+ adduct ion with cation of QSPIP^{R1}. This result further confirmed the synthesis of QSPIP.

Fig. R10. The mass spectroscopy of QSPIP.

Reference:

[R1] Kermit K. Murray, Robert K. Boyd, Marcos N. Eberlin, G. John Langlely, Liang Li & Yasuhide Naito. Definitions of terms relating to mass spectrometry (IUPAC Recommendations 2013). *Pure. Appl. Chem.* **85**, 1515-1609 (2013).

Revisions made: Fig. R10 was added as Supplementary Fig. 1.

Page 4, Line 7: A peak at $m/z = 156.15$ in mass spectroscopy was observed, which is consistent with the chemical structure of QSPIP ($C_8H_{18}N_3^+$, $M_w = 156$, Supplementary Fig. 1).

Page 18, Line 6: The QSPIP yield: 62.8%.

Page 18, Line 21: Mass spectroscopy (MS) was conducted on micrOTOF II (Bruker Switzerland) with electro spray ionization as ionization source.

2. At page 6, line 111, “It is noted that QSPIP is one of the rare strong electrolyte exploited as standalone monomers for straightforward preparation of nanofiltration membranes”. It is suggested to describe the advantages of QSPIP compared to other electrolytes that free QSPIP from the permanent limitations.

Response: Thanks a lot for this suggestion. In the revised manuscript we stressed two main advantages of QSPIP compared to other electrolytes. **1) Capability of straightforward interfacial polymerization.** Recently, many strong electrolytes were applied to modify polyamide membranes. The QSPIP is the first strong electrolyte, which served as standalone monomer to prepare defect-free Mg^{2+}/Li^+ nanofiltration membrane by straightforward interfacial polymerization. Due to the versatility of interfacial polymerization, in the revision we succeeded to prepare large-scale membranes ($2\ m^2$) and spiral wound membrane modules ($0.5\ m^2$), which also show reproducible and stable separation performance. **2) High chlorine-resistance.** The QSPIP monomer contain NO primary amine groups, thus the QSPIP-TMC membranes show good anti-chlorine ability due to the absence of chlorine-sensitive hydrogen in amide group ($-O=C-N-$).

Revisions made:

Page 17, Line 13: Collectively, this work shows that the QSPIP represents a novel design of monomer which combines high Mg^{2+}/Li^{+} separation performance, chlorine-resistance and scalability in one single membrane.

3. At page 7, line 139, “The orange and gray colors indicate section voids and inner voids, respectively”. The gray color indicates the occupied space of polymer skeleton that might not belong to free volume.

Response: Thanks a lot. Please see below the revised description in Fig. 2d, e.

Fig. R11. Molecular dynamics simulation of free volume of (a) QSPIP-TMC and (b) PIP-TMC networks, respectively.

Revisions made: Fig. 2d, e were replaced by Fig. R11a, b.

Page 7, Line 6: The orange and gray color indicate the voids of polymer and occupied space of polymer skeleton.

4. In page 6, line 120-125, it demonstrates that the spiral configuration of QSPIP increases the free volume of the polymer and thus enhances the microporosity. It is suggested to further compare QSPIP-TMC with BAPP-TMC, since BAPP has similar sizes but different molecular configuration?

Response: Thanks a lot. As shown in the added data (Fig. R12), the fractional free volume of BAPP-TMC is 12.3%, which is half of that of QSPIP-TMC (24.0%). This is due to the non-spiral configuration of BAPP monomer. Thus the permeance of BAPP-TMC ($2.4 L m^{-2} h^{-1} bar^{-1}$) is lower than that of QSPIP-TMC membrane ($23.0 L m^{-2} h^{-1} bar^{-1}$).

Fig. R12. Molecular dynamics simulation cell of free volume of BAPP-TMC.

Revisions made: Fig. R12 was added as Supplementary Fig. 10.

Supplementary Information: Page 9, Line 2: The fractional free volume of BAPP-TMC is 12.3%, which is half that of QSPIP-TMC network (24.0%). This is due to the spiral configuration of QSPIP that impedes the inefficient packing of polymer chain, thus BAPP-TMC membrane shows lower permeance ($2.4 \text{ L m}^{-2} \text{ h}^{-1} \text{ bar}^{-1}$) than QSPIP-TMC ($23.0 \text{ L m}^{-2} \text{ h}^{-1} \text{ bar}^{-1}$).

5. Fig. 4 presents the difference in the angles of water passage through the QSPIP and PIP, where the water has two transport angles for QSPIP because of the nonplanar conformation and strong interaction of quaternary ammonium and water. This result is interesting, but it is commonly considered that the transport channels is the voids between the polymer skeleton. Whether the authors propose that the rings of piperazine can also act as transport channels?

Response: Thanks a lot. It's true that water molecules permeate across membrane through the voids of polymer skeleton. The voids was surrounded by QSPIP and TMC segments and water molecules will pass through these segments. We meant to show that additional space (*i.e.*, free volume) were created due to the non-planar conformation of QSPIP, so that water molecules could pass the membrane *via* these volume (Fig. R13). We do not mean that water could pass across the piperazine ring directly. Both the schematic and discussion of water transportation through QSPIP and PIP (Fig. 4d) was revised. Please see below.

Fig. R13. Schematic of water transportation through QSPIP and PIP.

Revisions made: Fig. 4d was replaced by Fig. R13.

Page 10, Line 22: As schemed in Fig. 4d, more voids were produced by QSPIP in QSPIP-TMC membrane compared with PIP-TMC due to the spiral conformation of QSPIP. Water molecules permeate across QSPIP-TMC membrane *via* both the piperazine plane and region of quaternary ammonium, while water permeate across PIP-TMC membrane mainly *via* piperazine plane.

6. This work has proved the high stability of QSPIP-TMC in 200 ppm NaClO solution. Could the QSPIP-TMC endure higher NaClO concentration?

Response: Thanks a lot. As shown in the new data (Fig. R14), with increasing concentration of NaClO from 100 to 800 ppm, the flux of QSPIP-TMC increases slightly, while the R_{MgCl_2} keeps stable above 91%. Meanwhile, the surface morphology of QSPIP-TMC is smooth after being

treated by 800 ppm NaClO solution for 48 h and consistent with that of pristine QSPIP-TMC membrane (Fig. R14b, c). Thus the membrane could endure higher NaClO concentration.

Fig. R14. (a) Effect of NaClO concentration in the separation performance of QSPIP-TMC membrane. (Immersion time: 48 h, test conditions: 1000 ppm MgCl₂, 6 bar). Surface morphologies of QSPIP-TMC membrane (b) before and (c) after being treated by 800 ppm NaClO for 48 h.

Revisions made: Fig. R14a, c was added as Supplementary Fig. 14.

Page 11, Line 12: With further increasing NaClO concentration to 800 ppm, the R_{MgCl_2} of QSPIP-TMC membrane maintains high (>91%), while its flux increases slightly (Supplementary Fig. 14).

7. Please supply the samples preparation method for BET measurement in method section.

Response: Thanks a lot. The preparation method for BET measurement was previously shown in section 1.4 in supplementary information. Now we moved it to the method part in the main text.

Revisions made:

Page 18, Line 15: Preparation of QSPIP/PIP-TMC polymer (for BET measurements in Fig. 2f). TMC hexane solution (0.1 wt%, 400 mL) was mixed with QSPIP (or PIP) aqueous solution (0.5 wt%, 400 mL, pH=11), and the mixture was vigorously stirred for 5 min. The precipitation was filtered, washed with hexane for 3 times, water for 3 times, ethanol for 3 times and water for 3 times. The product was freeze-dried and stored for BET measurement.

REVIEWERS' COMMENTS

Reviewer #1 (Remarks to the Author):

Authors added new data that addressed reviewers' concern. The revised work will appeal to the general field and I support its publication.

[Note from the editor: reviewer #2 was not able to review the revised version of the manuscript and reviewer #1 was asked to look also over the response given to reviewer #2, see comments below.]

In the last round the #2 reviewer acknowledged the new chemistry, high performance, originality, etc, of the submission. And he/she raised concerns about the practical value and general interest on the work, without specific criticism. I evaluated the revised work, and found that the authors gave a convincing response to the reviewer's concern. They added a new figure, and prepared large membranes and modules to strength the practical value and general interest of their work. By using the membrane module, the authors managed to produce high purity Li_2CO_3 from simulated brine. They also explained the why the work appeal to broad readership. I think these added results are new compared to literature, and believe that the #2reviewer's concern was adequately addressed, and the work is suited for publication.

Response to reviewers

Reviewer #1 (Remarks to the Author):

Authors added new data that addressed reviewers' concern. The revised work will appeal to the general field and I support its publication.

Response: Thanks a lot.

[Note from the editor: reviewer #2 was not able to review the revised version of the manuscript and reviewer #1 was asked to look also over the response given to reviewer #2, see comments below.]

In the last round the #2 reviewer acknowledged the new chemistry, high performance, originality, etc, of the submission. And he/she raised concerns about the practical value and general interest on the work, without specific criticism. I evaluated the revised work, and found that the authors gave a convincing response to the reviewer's concern. They added a new figure, and prepared large membranes and modules to strength the practical value and general interest of their work. By using the membrane module, the authors managed to produce high purity Li_2CO_3 from simulated brine. They also explained the why the work appeal to broad readership. I think these added results are new compared to literature, and believe that the #2reviewer's concern was adequately addressed, and the work is suited for publication.

Response: Thanks a lot.